# Flexible Self-Powered Low-Decibel Voice Recognition Mask

**DOI:** 10.3390/s24103007

**Published:** 2024-05-09

**Authors:** Jianing Li, Yating Shi, Jianfeng Chen, Qiaoling Huang, Meidan Ye, Wenxi Guo

**Affiliations:** 1Department of Physics, College of Physical Science and Technology, Research Institution for Biomimetics and Soft Matter, Xiamen University, Xiamen 361005, China; 19820211153729@stu.xmu.edu.cn (J.L.); 19820180155642@stu.xmu.edu.cn (Y.S.); 19820211153720@stu.xmu.edu.cn (J.C.); qlhuang@xmu.edu.cn (Q.H.); mdye@xmu.edu.cn (M.Y.); 2Jiujiang Research Institute, Xiamen University, Jiujiang 332000, China

**Keywords:** triboelectric nanogenerators, vibration sensors, silent communication, speech recognition, human-computer interaction

## Abstract

In environments where silent communication is essential, such as libraries and conference rooms, the need for a discreet means of interaction is paramount. Here, we present a single-electrode, contact-separated triboelectric nanogenerator (CS-TENG) characterized by robust high-frequency sensing capabilities and long-term stability. Integrating this TENG onto the inner surface of a mask allows for the capture of conversational speech signals through airflow vibrations, generating a comprehensive dataset. Employing advanced signal processing techniques, including short-time Fourier transform (STFT), Mel-frequency cepstral coefficients (MFCC), and deep learning neural networks, facilitates the accurate identification of speaker content and verification of their identity. The accuracy rates for each category of vocabulary and identity recognition exceed 92% and 90%, respectively. This system represents a pivotal advancement in facilitating secure and efficient unobtrusive communication in quiet settings, with promising implications for smart home applications, virtual assistant technology, and potential deployment in security and confidentiality-sensitive contexts.

## 1. Introduction

Vibrations represent a pervasive phenomenon in a practical context. The triboelectric nanogenerator (TENG) provides a compelling method for generating electrical current through mechanical friction, presenting opportunities for charge transfer within materials without requiring additional power sources. Through the conversion of vibrations into electrical signals, the TENG has found diverse applications in large buildings, energy harvesting, mechanical equipment, health monitoring, and motion sensing [1,2,3,4,5,6,7,8,9,10,11,12,13,14,15,16,17,18,19,20,21,22,23,24,25,26]. As science and technology progress, the TENG’s role in signal sensing via vibration has expanded into speech recognition, prompting the need for improved material structure, response speed, sensitivity, noise immunity, and deployment contexts. Notably, its capacity to detect signals through vibration has significantly contributed to advancements in acoustic science [27,28,29,30,31,32,33,34,35,36,37]. For instance, Zheng’s team engineered a TENG capable of recognizing road and traffic sounds and deployed it on roads or buildings to differentiate between passing vehicles and structural flaws [38]. Similarly, Wang et al. coated silk protein onto the copper film of the device and combined it with a layer of FEP coated with silver, forming upper and lower layers. This setup allows for the detection of sounds within the frequency range of 20 to 2000 Hz and exhibits good noise resistance [39]. Furthermore, Xia et al. developed a TENG adept at detecting minute pressures for monitoring breath and glottal vibration articulation using graphene oxide as the raw material [40]. Wang’s team fabricated layered thermoplastic polyurethane (PU) and ionic electrode layers atop a polytetrafluoroethylene film, enabling the detection of minute vibrations from acoustic pressures and the identification of diverse human voice signals [41]. In recent years, with the development of artificial intelligence technology, there have been corresponding breakthroughs in TENG in the field of speech recognition. Appendix A lists the progress of TENG in speech recognition applications.

Most of these studies have made significant progress in voiced speech recognition while contributing less to silent speech recognition. Additionally, integrating voiceprint recognition into silent speech recognition designs is necessary for system security. The integration of voice communication with modern technologies such as smart homes and voice assistants is also a growing trend. Therefore, there is an urgent need to further explore the expansion of TENG in speech recognition. The system to facilitate communication in environments necessitates careful attention to materials, precise speech recognition technology, and compatibility across various life aspects. The characteristics of TENG, such as self-powering capabilities, high sensitivity, and efficiency, facilitate the applications of intelligent sensing technologies in our daily lives. For instance, integrating TENG into gesture recognition [42,43,44], intelligent IoT [45,46,47], wearable devices [48,49,50], automated monitoring [51,52,53], and speech recognition [39,54,55,56]. Despite significant advancements in speech recognition technology, challenges persist in certain specific application scenarios regarding accuracy and stability. For example, challenges include speech recognition in noisy environments and the collection and processing of long-duration speech signals. Friction-based nanogenerators may offer a novel approach to addressing these challenges, potentially providing new possibilities for self-powered systems, particularly in scenarios requiring long-term operation without easy battery replacement, such as remote monitoring devices or wireless sensor networks. Furthermore, integrating speech recognition technology with friction-based nanogenerators can enable more intelligent self-powered systems, such as utilizing ambient sound signals to activate or control devices, thereby achieving more intelligent and adaptive functionality. However, the combination of friction-based nanogenerators and speech recognition also faces several challenges. Firstly, the energy conversion efficiency of friction-based nanogenerators is relatively low and needs further improvement to meet practical application demands. Secondly, the application scenarios are limited, necessitating the expansion of the functionalities of friction-based nanogenerators and the design of more stable and diversified nanogenerator systems. Additionally, addressing integration and optimization issues between friction-based nanogenerators and speech recognition systems is necessary to achieve good coordination and collaboration between the two.

Overall, the combination of friction-based nanogenerators and speech recognition holds tremendous potential, offering new solutions and applications for self-powered systems, the Internet of Things (IoT), and beyond. However, it is currently in the research exploration stage, requiring further engineering and scientific research to address technical and application challenges and achieve widespread application in real-world scenarios.

Here, we developed a triboelectric nanogenerator (TENG) tailored for silent speech recognition, as depicted in Figure 1a. Our device design entailed engineering a single-electrode, contact-separated TENG (CS-TENG) by selecting the contact-separated operation mode from the four TENG modes [57]. The contact-separated structure allows for free movement of a charged electrode, where periodic changes in the contact area between surfaces facilitate charge transfer and electrostatic induction. This in turn drives electron flow in external circuits and expands TENG’s applications across diverse contexts [58]. Practically, we affixed the CS-TENG to a mouthpiece to capture aeroacoustics signals and simulate a silent communication environment such as whispering and silent reading. Through the application of filtering, data processing, and deep learning algorithms, we have successfully achieved speech recognition. When integrated into scenarios such as smart homes and voice assistants, this approach simultaneously presents a novel solution for the silent operation of intelligent technology. Additionally, we bolster the system’s security based on a unique personal voiceprint feature mode to authenticate user’s identities.

## 2. Materials and Methods

The device structure consists of silk film, conductive fabric, and Polytetrafluoroethylene (PTFE), as shown in Figure 1b. PTFE and conductive fabric served as the friction electrodes. The two electrodes are then encapsulated with a treated soft silk fibroin (SF) film, which has good biocompatibility and mechanical qualities. The top and lower electrodes are separated and contacted when the CS-TENG vibrates. Owing to the materials’ different electronegativities, the mechanical energy from detected vibrations is transformed into electrical signals during contact separation. Subsequently, as shown in the flowchart of Figure 1c, TENG generates output signals for data processing by machine learning algorithms to establish a speech recognition system.

Figure 2a demonstrates that pure SF films exhibit poor tensile properties, breaking under a stress of 18 MPa at only 1.8% strain. In contrast, PU films have good tensile properties [59]. To improve the tensile properties of pure-SF films, the addition of PU leads to PU-modified SF films (treated-SF) exhibiting enhanced tensile strength over pure-SF films, achieving a strain of 120% and a stress of 23 MPa. The water loss rate of the treated SF film exceeded that of polyethylene (PE) film, approximately 5.2 mL h^−1^, demonstrating the silk film’s superior air permeability (Figure 2b). As depicted in Figure 2c,d, the biocompatibility was assessed using the CCK-8 cell counting method to evaluate cell viability and proliferation on the treated-SF films after one week of incubation. The result shows that the cell density on the samples increased significantly, and their surfaces became progressively covered with cells. The findings indicate that the cells demonstrated robust adhesion, spreading, and growth on the treated SF. This experimental result demonstrates the excellent biocompatibility of the modified silk membrane [55,60].

## 3. Results and Discussion

### 3.1. Performance of CS-TENG

As depicted in Figure 3a, after curve fitting, the sensitivity of CS-TENG reaches 7.43 × 10^−2^ nA kPa^−1^ under 44 kPa and decreases to 6.52 × 10^−3^ nA kPa^−1^ within 44 to 222 kPa. The rapid linear current increase at low pressure results from an accelerated expansion in the contact area between the upper and lower friction electrodes (conductive fabric and PTFE) upon external force application. This illustrates that the device’s sensitivity at low pressure significantly exceeds that at medium and high pressure, making it suitable for some applications like small vibration detection or whispered speech communication. Concurrently, the swift reduction in sensitivity at medium and high pressures broadens the sensor’s detection range, enabling sensitive and extensive dynamic pressure monitoring. To determine the optimal size and area for the TENG, we conducted tests across various dimensions to observe signal changes when the device was tapped by a motor at a consistent frequency. The comparison reveals an increase in device current with size (Appendix A), likely due to the expanding device area increase in the contact area between the conductive fabric and PTFE. The current rises with the thickness increment from 1 mm to 4 mm; however, beyond 3 mm, the increase in current becomes less pronounced (Appendix A). Here, a device measuring 2 cm × 2 cm with a 2 mm thickness was consistently selected for testing.

For CS-TENG to stably detect sound signals, it necessitates precise, stable, and repeatable vibration signal sensing across a broad frequency spectrum. CS-TENG accurately detects every high-frequency vibration (40 to 1020 Hz), as evidenced by fast Fourier transform (FFT) analysis (Figure 3b,c). The FFT spectrogram revealed that the response signal’s frequency shift compared to the original frequency resulted in a frequency deviation of no more than 0.63%. This demonstrates the device’s heightened accuracy in high-frequency vibration sensing with little signal distortion. By amplifying the shaker’s vibration intensity at a steady vibration frequency (100 Hz), the sensor’s current signal incrementally rose from 1 nA to 12 nA. This signifies that the current magnitude precisely mirrors the actual vibration intensity. Figure 3d depicts a fitting of the relationship between current output and vibration intensity, approximating a proportional curve. Figure 3e,f illustrates the CS-TENG vibration sensor’s current output under continuous scanning mode across frequencies ranging from 10 to 1020 Hz. The initial current amplitude is significantly higher than that at the end, demonstrating CS-TENG’s enhanced sensitivity to vibrations below 200 Hz. Human vocal frequencies normally occur within 50 to 200 Hz. Consequently, CS-TENG can effectively detect the vibrations associated with human vocalizations. To assess stability under low-frequency vibration, electrical signals were measured in 0.5 Hz intervals from 0.5 Hz to 3 Hz (Figure 3g,h). The test results indicated a modest increase in voltage signal magnitude with ascending frequency. The current amplitude experienced a more pronounced increase as the vibration frequency rose. Consequently, the current signal demonstrated greater sensitivity to vibration perception in low-frequency tests, meaning its output was more significant at identical vibration frequencies. In later voice vibration signal acquisitions, the analysis mainly relies on the current signal. To evaluate its repeatability and durability, voltage and current signals were measured after 30 days. We found that signals are closely matched to those recorded on the first day (Figure 3i), ensuring the TENG works soundly for long-term sound signal acquisition.

### 3.2. Speech Recognition Function of CS-TENG

Public spaces necessitate a degree of quiet, with studies indicating that noise levels between 20 and 40 dB are conducive to human activity and rest. In the current state of intelligent technology, silent/low-decibel speech recognition still requires further improvement and dissemination. Appendix A presents a comparison and analysis of current speech technologies. In this study, the CS-TENG we developed generates an electrical signal by facilitating contact separation between its upper and lower stages through aeroacoustics transfer airflow induced by whispering. To verify the speech perception capability of CS-TENG, we established a testing mechanism where CS-TENG is attached to a mouthpiece in a laboratory setting (52 dB ambient noise). By speaking at a soft air sound level under ambient noise, the TENG is induced to vibrate, allowing for the detection of the output current signal and the observation of waveforms. The results are then compared and analyzed, and the system is applied in various speech-control scenarios. Figure 4a illustrates the application schematic for this system. On the left are our physical figures. After signal acquisition, the processed signals are input into the model for recognition, and the recognition results are applied to applications such as smart homes and voice assistants. This chapter will elaborate on the processes of signal processing and speech recognition.

Initially, we uttered the same word at a consistent decibel level across varying distances directly in front of the TENG, observing that the output becomes more distinct and the signal change magnitude increases as the distance to the mouthpiece decreases. At 8 cm, the signal is weak. At 6 cm, it is the clearest. And beginning at 5 cm, although the signal strengthens, breathing starts to interfere with the speech signal (Appendix A). Consequently, we selected 6 cm as the optimal testing distance. Subsequently, we further investigated the characteristics of speech signals using a tester. Words of varied pronunciations were classified into three categories: commands, functions, and home. Furthermore, signals from seven distinct sentences were gathered. Figure 4b displays the real-time current signals detected by CS-TENG when a volunteer articulates words and sentences across the three categories (commands, functions, and home). Figure 4c depicts the process of filtering the original signal, followed by the application of the short-time Fourier transform (STFT), revealing three signals in the spectrogram. These signals correspond to three segments of the time-domain waveform, averaging a frequency range between 50 and 75 Hz.

Following the conversion of the signal from the time domain to the frequency domain by STFT, we utilized Mel-frequency cepstral coefficients (MFCC) to extract features from pertinent sound signals in the frequency domain and then trained these features using a neural network model. Following parameter optimization, the model can swiftly learn from the signal features and categorize them into specific groups. In this study, the training model is a neural network that combines convolutional neural networks (CNN) and long short-term memory (LSTM). Initially, the CNN is utilized for advanced feature extraction, isolating valid sound signals, and diminishing the original data volume. Subsequently, the LSTM neural network is selected to optimally process the output of the feature by the CNN, assimilate prolonged effective information, and facilitate its training and classification. Analyzing speech waveforms across three-word categories—commands, functions, and home—and seven common sentences, the recognition accuracies were 92.4%, 95.7%, 96.2%, and 88.2%, respectively. The corresponding confusion matrices for the word recognition accuracy are shown in Appendix A. Figure 4d shows the confusion matrix for sentence recognition accuracy. The combined classification of instructions, functions, and home encompasses a total of 29 terms, achieving an accuracy of 89% (Appendix A). Here, an analysis of the recognition results has been conducted. The accuracy of recognition may be influenced by factors such as the testing environment and friction charges on the device. Therefore, to further increase accuracy, it is important to ensure consistency in signal collection and result verification environments. The accuracy of sentences may be slightly lower compared to individual words, possibly due to the fact that sentences are composed of different words, and there may be slight variations in pauses and pronunciation during each signal collection. When consolidating the three categories of words, the accuracy decreases. This could be because more diverse classifications require larger datasets for support. Therefore, to improve the accuracy of mixed vocabulary, extensive data collection is necessary, followed by multiple rounds of training.

Furthermore, traditional machine learning algorithms, including support vector machine (SVM), random forest (RF), decision tree (DT), and logistic regression (LR), were compared, with their accuracies detailed in Figure 4e. Overall, LR performed best and the DT algorithm the worst. However, all traditional models underperformed compared to the LSTM neural network mode. Our recognition outcomes are presented in a QT-developed GUI interface, which displays the speech’s electrical signal waveform and the recognition results; the interface is depicted in Figure 4f. The signal waveforms illustrating the pronunciation of common words in daily life are shown in Appendix A.

### 3.3. Identification and Security Defenses

In daily life, security system applications encompass a broad range, including mobile phone unlocking, private robot control, smart homes, and smart vehicles. Consequently, it is crucial to design a security system based on voice recognition. One advantageous method to enhance the security of speech systems is to utilize biometric features for encryption. Biometric features refer to certain physiological or behavioral characteristics unique to each individual that can be effectively measured, identified, and verified through technology, such as facial features, fingerprints, and voiceprints. Biometric recognition technology utilizes various sensing and data collection techniques, as well as artificial intelligence techniques like deep learning, to calculate individual physiological and behavioral features, establish mathematical models, and perform recognition and differentiation. In recent years, there has been a growing demand for biometric recognition technology in various fields. Among various biometric recognition technologies, intelligent speech-based technology is one of the most widely applied.

Figure 5a displays the speech waveforms of four volunteers. For the word “computer”, both the positive and negative amplitudes, along with the mean and extreme values of signal strength for each volunteer, were calculated. It was observed that the signal characteristics varied significantly among individuals (Figure 5c). Subsequently, the speech correlation among different users was further analyzed. Pearson’s coefficients for the four volunteers who articulated the word “computer” were computed based on feature parameters to assess the speech-signal correlation among different users. Coefficients closer to 1 indicated a stronger correlation, while those closer to −1 indicated a weaker correlation. Appendix A reveals that the correlation between speakers and their signals approaches 1, indicating strong self-consistency, whereas correlations with signals from other testers are significantly lower, highlighting substantial individual variability in articulatory signals. This indicates that, despite volunteers having distinct voices, each possesses a unique speech pattern. Leveraging this, a personalized speech recognition system can be tailored to everyone, establishing a one-to-one match between their vibration signals and spoken words. Consequently, personalized model training, tailored to the unique spectral properties of each user’s voice signals, can accurately identify the four users, serving as a robust authentication mechanism for security. In this study, 10 distinct vocabulary sets comprising a total of 100 words were compiled into a dataset, achieving a recognition accuracy of 90.6%, with the confusion matrix depicted in Figure 5d. By training the developed classifier, the system can identify specific users. Upon a user inputting a unique voice password, the system determines if they are a registered user, thereby enhancing the smart device’s security. The judgment result is additionally presented in the GUI interface (Figure 5e).

## 4. Conclusions

In this study, we developed a novel device named CS-TENG and applied it in the field of acoustics. CS-TENG exhibits excellent stability and accuracy in both low- and high-frequency ranges, rendering it highly promising for applications in acoustic signal processing. Specifically, our research team utilized CS-TENG to detect vibrations produced by whispered speech and successfully recognized spoken content by analyzing the waveforms and spectrograms of these signals. This technology not only demonstrates good stability and accuracy in both low and high-frequency ranges but also enables the possibility of controlling smart technology through voice commands in quiet environments without causing disturbance, thus offering great flexibility for practical applications in various environments and scenarios.

Here, the feasibility and effectiveness of this technology were validated through experiments. We identified common words and sentences in daily life and applied them to voice assistants and smart home systems. In the experiments, we improved recognition accuracy through neural network algorithms, further enhancing system performance. These experimental results indicate that this technology not only performs well in laboratory environments but also holds potential for practical applications. In addition to its application in speech recognition, we also explored the potential application of this technology in user identity recognition. By analyzing the speaker’s voice characteristics, we can identify different individuals, thereby achieving personalized voice interaction experiences. This application of user identity recognition can enhance security defense and protect personal privacy information within speech systems, opening up more scenarios for the application of speech systems.

In summary, this study provides a new approach and method for the development and application of speech recognition technology. By utilizing CS-TENG’s vibration signals for speech recognition, we not only improve recognition accuracy and stability but also address some issues existing in traditional speech recognition systems. This technology has broad application prospects and is expected to play a significant role in human-computer interaction and communication in quiet places such as libraries and meeting rooms, as well as in information protection and security systems. In the future, with further research and improvements in this technology, it will bring new development opportunities for smart technology and human-computer interaction, providing more convenience and security for people’s lives.

## 5. Experimental Methods

Experimental Design for the treated-SF: Following the established protocol for silk solution preparation [61,62,63], the silk solution underwent degumming, washing, drying, dissolving, and centrifugation. Subsequently, the PU solution was diluted to a ratio of 1:13 (PU: water, *v*/*v*). A mixture of silk solution (1 mL) with PU solution (9 mL) was homogeneously blended and gradually transferred into a petri dish, which was then incubated at 26 °C for 48 h to yield a film.

Experimental Design of CS-TENG: PTFE and conductive fabrics were selected as the friction electrodes. The conductive fabrics, crafted from high-strength polyvinyl acetate (PVAC) fiber cloth, are plated with copper and nickel metals to enhance conductivity and coated with a conductive acrylic adhesive. Wires are attached to the conductive fabric. Following this, both the upper and lower electrodes are encapsulated within treated SF. A gap between the upper and lower electrodes is maintained and sealed around its perimeter with nanoglue to enable the contact separation of the electrodes.

## Figures and Tables

**Figure 1 sensors-24-03007-f001:**
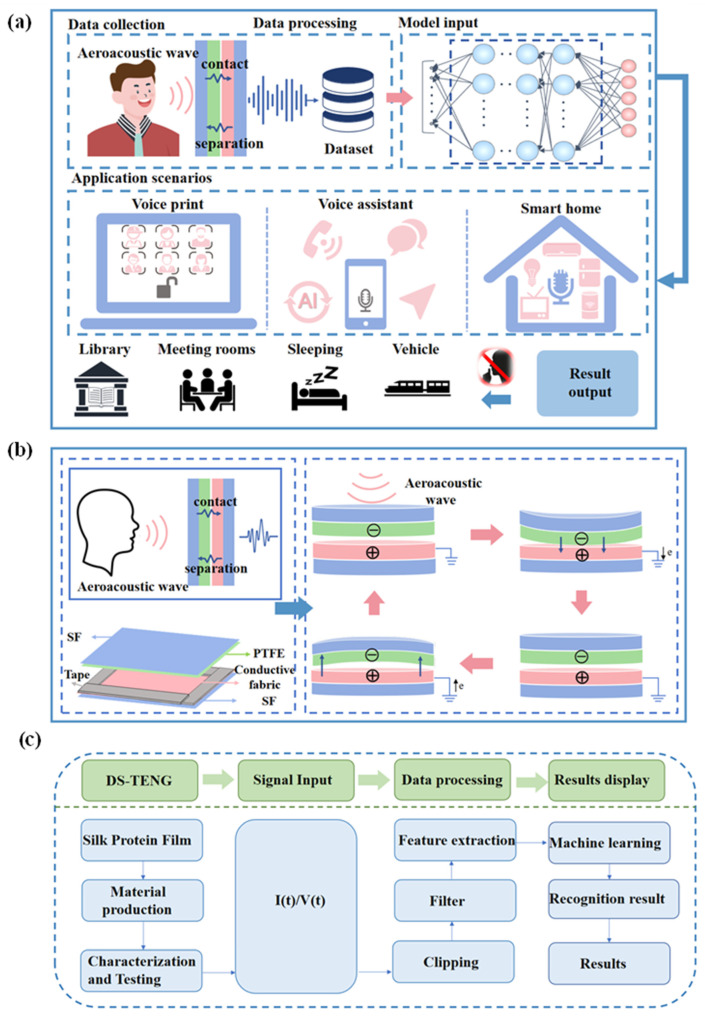
Overall schematic and flowchart: (**a**) Schematic diagram of the structure and application scenarios of CS-TENG; (**b**) Working mechanism of CS-TENG; (**c**) Flowchart of the speech recognition system.

**Figure 2 sensors-24-03007-f002:**
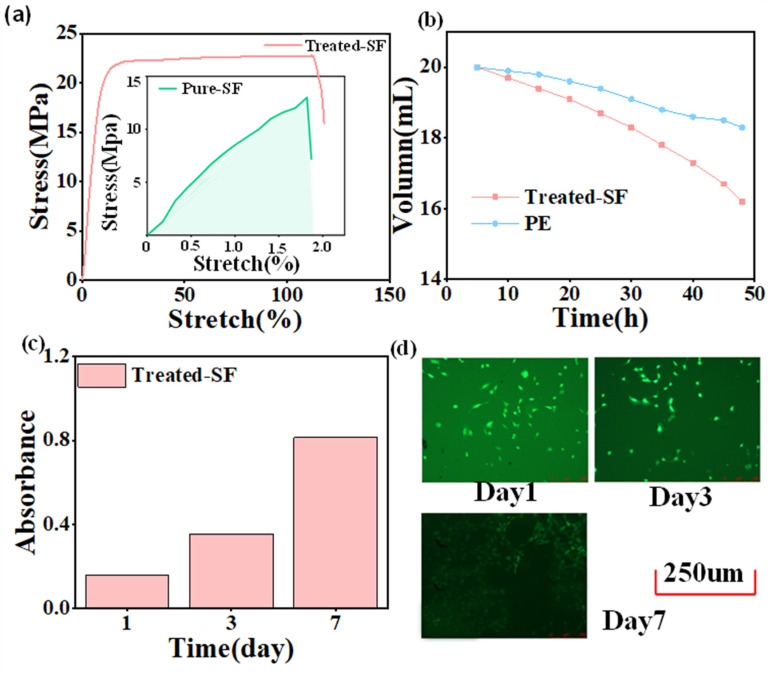
Characterizations of the treated SF: (**a**) Comparison of tensile properties between treated SF and pure SF; (**b**) Permeability test; (**c**) Biocompatible cell counting; (**d**) Cell fluorescence map (scale bar: 250 μm).

**Figure 3 sensors-24-03007-f003:**
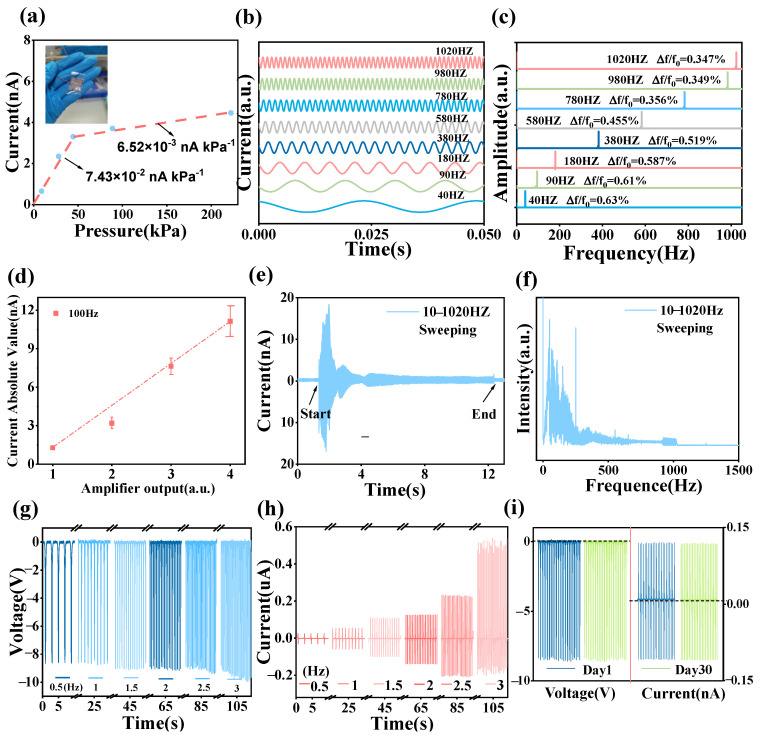
Device performance tests: (**a**) Current sensitivity measurements; (**b**,**c**) High-frequency vibration test performance and its corresponding FFT; (**d**) Variation in current with vibration intensity; (**e**,**f**) Sweep signal and its frequency domain FFT; (**g**,**h**) Low-frequency vibration signals in voltage or current; (**i**) Working repeatability and durability after 30 days.

**Figure 4 sensors-24-03007-f004:**
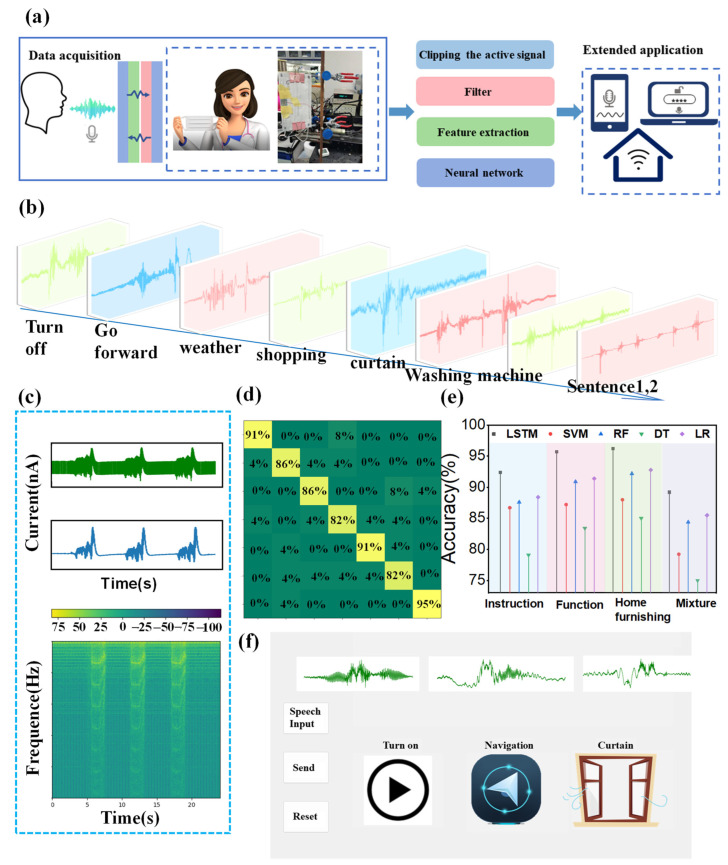
Acquisition of speech signals: (**a**) Schematic of the flow of speech signal acquisition and recognition; (**b**) Current waveform acquisition of the current of speech signals for different kinds of words (command, function, home) and sentences; (**c**) STFT after noise filtering of raw speech signals; (**d**) Confusion matrix of sentence recognition results; (**e**) Comparison of the accuracy of different algorithms for recognizing three categories of words (instruction, function, home); (**f**) Display interface of recognition results.

**Figure 5 sensors-24-03007-f005:**
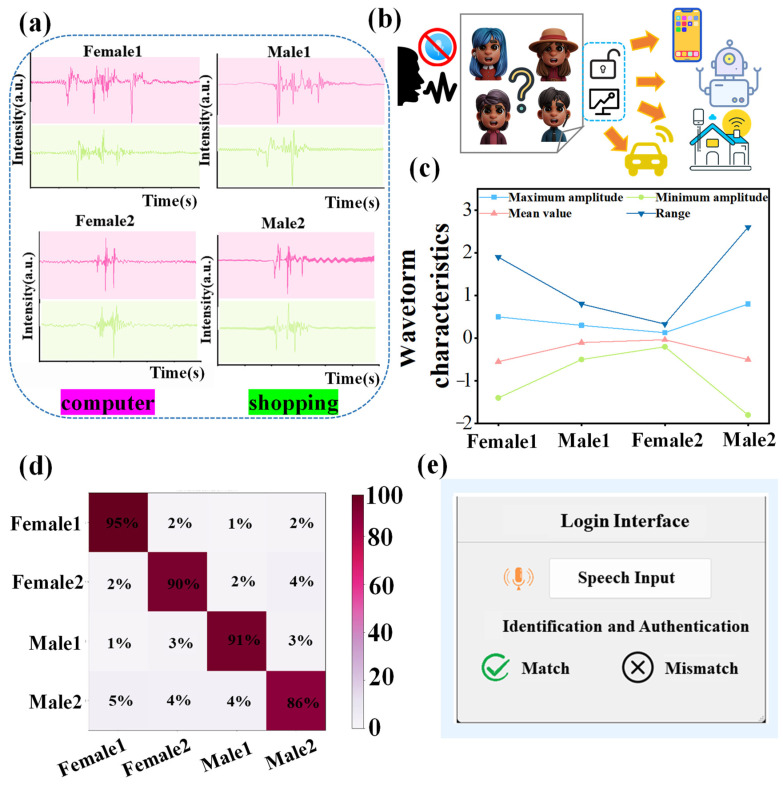
Recognition and analysis of voice signals: (**a**) Speech signal waveform of four testers speaking the same word; (**b**) Application scenario of voiceprint recognition; (**c**) Comparison of signal eigenvalues for different testers; (**d**) Confusion matrix for identity recognition accuracy; (**e**) Display interface for identity recognition.

## Data Availability

Data are contained within the article and Appendix A.

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
