# Peer review of "Flexible Self-Powered Low-Decibel Voice Recognition Mask"

_sensors, 2024, doi:10.3390/s24103007_

Round 1
Reviewer 1 Report
Comments and Suggestions for Authors
The study introduces a novel application of triboelectric nanogenerators in voice recognition; however, it requires significant enhancement in terms of experimental detail, data analysis, practical application considerations, and theoretical underpinnings. Additionally, the manuscript would benefit from a thorough language review to ensure clarity and precision. A major revision will allow the authors to address these concerns comprehensively, thereby strengthening the contribution of their research to the field.
1)While the integration of a triboelectric nanogenerator (TENG) into a voice recognition mask is an interesting approach, the paper could benefit from a more rigorous discussion on the innovation compared to existing solutions. The authors should clarify how their device outperforms or differs significantly from current noise-cancellation and voice recognition technologies.
2)The experimental section provides a general overview of the device's construction and testing. However, the lack of detailed experimental protocols, such as specific conditions for the tests (e.g., humidity, temperature), could leave room for questions regarding the reproducibility of the results. The authors should include comprehensive details to allow for independent validation of their findings.
3)The results section should include a critical analysis of the data, including the limitations of the current design and potential sources of variability. The authors should discuss any discrepancies between expected and observed results and how these were addressed.
4)The paper would benefit from a more thorough graphical representation of the data. For instance, graphical comparison of the CS-TENG's performance metrics with those of other voice recognition technologies could provide a clearer picture of its advantages and disadvantages.
Author Response
Please see the attached response document

Reviewer 2 Report
Comments and Suggestions for Authors
The manuscript is well written with detailed data analysis and results with s scope for relevant applications.
1. What is the basis for the author’s claim that PU-modified SF films are biocompatible? Is it the evidence from Figure C – biocompatible cell counting or any other literature reference?
2. The authors in reference 41 have used similar materials with PU and PTFE configurations to fabricate the sensors for their measurements still achieving the same quality. How does the present work advance the materials fabrication other than adding biocompatible, treated SF and how does this material improve the quality of measurement or sensor?
3. Is there any specific reason for not including different age groups of individuals or non-native English speakers for recognition and voice analysis?
4. How do you list and define the characteristics or unique features of these sensors?
5. The material fabrication adopted by the authors has a lot of similarities with some of the recent articles from the literature i.e. Chaoran Liu, Yishao Wang, Nan Zhang, Xun Yang, Zuankai Wang, Libo Zhao, Weihuang Yang, Linxi Dong, Lufeng Che, Gaofeng Wang, Xiaofeng Zhou, A self-powered and high sensitivity acceleration sensor with V-Q-a model based on triboelectric nanogenerators (TENGs), Nano Energy, Volume 67,2020,104228, ISSN 2211-2855, https://doi.org/10.1016/j.nanoen.2019.104228
Author Response

(The authors gave the same response as above.)

Reviewer 3 Report
Comments and Suggestions for Authors
this manuscript provides a new approach and method for the development and application of speech recognition technology. By utilizing CS-TENG's vibration signals for speech recognition. it is very interesting and There are several issues to be addressed before being published.
1. It is suggested that the authors can give some comparisons to previous work with this work and make a table or diagram.
2. What is the stability of device? authors has shown the Working repeatability and durability after 30 days,how about after more days?
3. there are writing errors. like um.
Author Response

(The authors gave the same response as above.)

Round 2
Reviewer 1 Report
Comments and Suggestions for Authors
All concerns addressed. Suggest accept as is.